# Adolescent sexual health interventions that include very young adolescents in sub-Saharan Africa: a scoping review protocol

Wanangwa Chimwaza Manda [1,2] Yandisa Sikweyiya,[3,4] Blessings Nyasilia Kaunda-Khangamwa,[1] Apatsa Selemani,[2] Scholastica Jimu,[5] Mphatso Kamndaya[6]

For numbered affiliations see end of article.

**Correspondence to**
Wanangwa Chimwaza Manda; wmanda@cartafrica.org

## ABSTRACT

**Introduction** Targeting very young adolescents (VYAs) with sexual health (SH) interventions is increasingly being recognised as one of the strategies for addressing SH challenges in late adolescence. However, there is a dearth of literature regarding SH interventions implemented specifically for VYAs in sub-Saharan Africa (SSA). This scoping review aims to provide a summary of documented evidence on SH interventions that include VYAs in SSA, identify gaps in existing interventions and provide recommendations for further programmatic work on SH for VYAs.

**Methods and analysis** The methods for this scoping review will be guided by the framework proposed by Arksey and O'Malley and further enhanced by Levac *et al* and the Joanna Briggs Institute. We will search electronic databases: Popline, EMBASE, PubMed, CINAHL, Dimensions, African Journals Online (AJOL) and specific summon country-specific search. We will include published studies from SSA and only adolescent SH interventions published from the year 2003–2022. Furthermore, we will include programmatic and intervention literature that has not been published in peer-reviewed articles. The data will be charted using the Preferred Reporting Items for Systematic Review and Meta-Analysis Extension for Scoping Review. The data will then be collated and summarised.

**Ethics and dissemination** The scoping review methodology involves putting together information from articles or grey literature that is either publicly available or shared by the authors, this study does not require ethical approval. Findings of this scoping review will be published in a scientific journal and presented at relevant scientific fora and conferences. This scoping review will provide a comprehensive overview of the evidence base of adolescent SH interventions for VYAs in SSA and will highlight critical gaps in the existing interventions and areas where further programmatic work is needed for VYAs in SSA.

**Registration** https://archive.org/details/osf-registrations-gn538-v1.

## BACKGROUND

The UNICEF defines adolescence as the period between the ages of 10 and 19.[1] Adolescents are assigned into two categories of older adolescents (ages 15 to 19) and very young adolescents (VYAs).[1] Globally, there are 1.2 billion adolescents, and half are between the ages of 10 and 14.[2] Low and middle-income countries (LMICs) inhabit about 90% of these adolescents.[3] During the period of adolescence, an individual undergoes rapid physical, biological and intellectual changes as he or she transitions from childhood to adulthood.[4 5] The period of adolescence is also regarded as the time in which the person advances from the onset of secondary sex features to that of sexual matureness, for example, increased facial, chest or pubic hair and upper body muscle for boys, while girls show rounded figure/hips, breasts budding and the onset of the menstrual cycle.[6] Furthermore, mental functions and self-identity patterns transition from those of a young child to those of an autonomous adult.[7] As the biological, cognitive,

### STRENGTHS AND LIMITATIONS OF THIS STUDY

⇒ The search strategy includes a range of electronic databases with peer-reviewed literature, including grey literature, Popline, EMBASE, PubMed, CINAHL, Dimensions, African Journals Online (AJOL), Google, Directory of Open Access Journals (DOAJ) and summon country-specific search.

⇒ Challenges are expected in finding published articles on very young adolescents (VYAs) sexual health (SH) interventions since research on VYAs has been limited until recent years.

⇒ Since we are focusing on interventions and programmes of which programme implementers are usually not researchers, it is highly likely that some SH interventions or programmes involving VYAs may not have been published in peer-reviewed articles.

⇒ Stakeholders will be consulted to provide insights in to interventions or programmes that are not published or may have been missed by the scoping review.

BMJ

psychosocial and emotional changes are happening, the adolescents also experience changes in comprehending life, their relationships, spaces and supports they interact with across adolescence.[6] In addition, it is during this period that new gender roles and sexual identities are established, various information is acquired, new attitudes are developed and new behaviours and relationships are tried.[8 9] These developments and experiences put adolescents at risk of various sexual health (SH) challenges, including infections that are transmitted sexually such as HIV, and early unintended pregnancies.[10–16]

The majority of VYA are not yet engaging in penetrative sex; however, their sexual curiosities have begun and some are reported to be involved in non-coital activities such as kissing, cuddling, foreplay and heavy petting.[2 17–19] In addition, few studies conducted in sub-Saharan Africa (SSA) suggest that some VYA boys and girls engage in unsafe heterosexual vaginal sex either willingly or forced.[20–23] Evidence from studies conducted in Malawi, Swaziland, Zimbabwe and Kenya indicates that many VYA boys and girls were coerced into sex the first time they had sexual intercourse.[24] These findings indicate that VYAs face various challenges related to their SH.

Most of the SH challenges occurring in early adolescence emanate from sexual risk behaviours that are linked to social-cultural factors, including gender norms and socialisation, socioeconomic factors and structural factors.[8 11–16 24–26] For instance, expectations for gender roles that young boys should experiment sexually to prove their manhood promote their perceived superiority over girls, sexual aggressive behaviour and sexual risk taking.[24 27] On the other hand, the social expectation that girls should be passive, disadvantage them by reducing their power and agency to reject unwanted sex or negotiate safe sex.[24 27] Recent studies in gender and education conducted in South Africa showcase how heterosexuality practices in primary schools promote oppressive behaviours against girls, including violence, sexual coercive and humiliation.[28 29] Furthermore, barriers to education such as school fees, long distances to school, cultural norms that promote boy's education only and unfavourable school environments where girls encounter violence may cause young girls to drop out of school, thereby increasing their likelihood of engaging risky sexual behaviours such as premarital sex and transactional sex.[24 27] In addition, poverty that results in economic difficulties has been shown to be a cause of child marriage, adolescent pregnancy, unsafe sex, sexual activity involving multiple concurrent partners and fewer educational possibilities.[11–15 24 27] These risky sexual practices have been shown to have detrimental effects on the sexual and reproductive health of VYA girls, leading to adolescent pregnancies, unsafe abortions and infections transmitted through sex such as HIV.

Every year, about 3.9 million girls aged between 15 and 19 years have unsafe abortions.[30] Around 2.1 million adolescents worldwide, including 770 000 VYA and 1.3 million older teenagers, were projected to have HIV in 2016.[31] In LMICs, it is projected that 2 million VYA girls and 21 million girls within the age range of 15 and 19 give birth each year.[32] Furthermore, In LMICs where most of the SH challenges among VYAs happen, the furtherance of adolescent sexual and reproductive health through putting more effort into positive youth development is increasing.[18 27 33 34] As such, adolescent sexual and reproductive health programmes are being implemented in various countries around the world through a variety of channels, such as peer education programmes, youth centres, youth-friendly health services and family life education.[9 27 35] The majority of adolescent SH programmes, however, neglect the SH needs of VYAs and instead concentrate on older adolescents.[2 18 27 36] Not only are VYAs overlooked in SH interventions, but there is also a dearth of literature regarding sexual and reproductive health programmes that include VYAs in SSA (SSA).[35 37] This scoping review aims to give a summary of documented evidence on SH interventions that include VYAs in SSA, identify gaps in existing interventions and provide recommendations for further programmatic work on SH for VYAs in SSA. Since targeting VYAs with SH interventions is increasingly being recognised as one of the solutions to tackling SH challenges in late adolescence,[8 27 38] it is envisaged that findings from this scoping review will contribute to the evidence that is required to develop evidence-based, culturally and age-appropriate primary prevention programmes and policies for addressing SH needs of VYA in SSA.

## Specific objectives

1. To describe the types of SH interventions that have been implemented for VYAs in SSA.
2. To describe the settings in which the SH interventions for VYAs have been implemented in SSA.
3. To document the barriers and facilitators of implementing SH interventions for VYAs in SSA.
4. To identify gaps in SH interventions implemented for VYAs in SSA.
5. To provide recommendations for further programmatic work on SH for VYAs in SSA.

## METHODS AND ANALYSIS

The framework suggested by Arksey and O'Malley, which has been improved by Levac *et al* and the Joanna Briggs

| Table 1 | Electronic search record | | |
|---|---|---|---|
| **Date** | **Keyword searched** | **Search engine used** | **Number of publications retrieved** |
| – | | | |

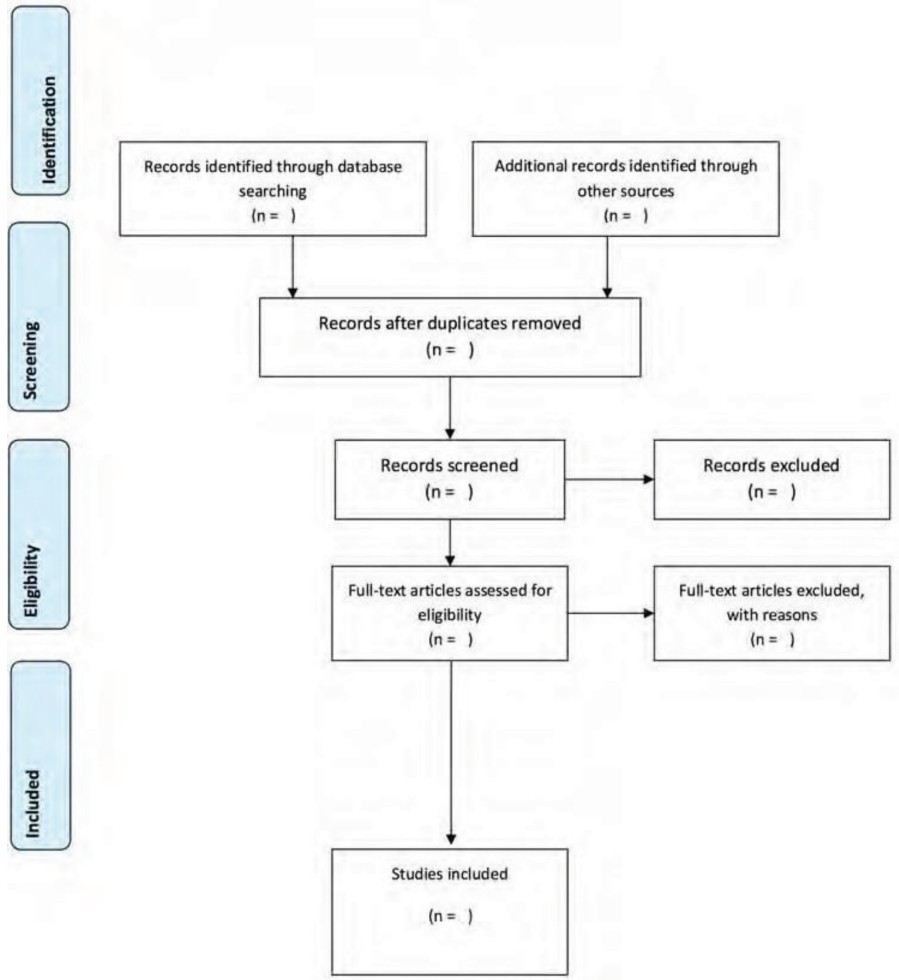

**Figure 1** Flow diagram describing the process of articles and publications being reviewed and selected. (1) Identification: articles will be identified through a database search and through searching of other sources, for example grey literature. (2) Screening: the identified records will be screened and duplicates removed. (3) Eligibility: full-text articles will be screened for eligibility and those not falling within the inclusion criteria will be excluded. (4) Included: studies that will finally be included for the review.

Institute, will serve as the basis for this scoping review's methodologies.[39 40] The stages in the framework include:
► Stage 1. Identifying the research question.
► Stage 2. Identifying relevant studies.
► Stage 3. Study selection.
► Stage 4. Charting the data.
► Stage 5. Collating, summarising and reporting the result.
► Stage 6. Consultative forum.

**Stage 1: identifying the research questions**
This scoping review has the following research questions:

**Main question**
What evidence is there for SH interventions that have been implemented for VYAs in SSA?

**Specific questions**
1. What types of SH interventions have been implemented for VYAs in SSA?
2. In what settings have these interventions been implemented?
3. What have been the implementation barriers and facilitators of these SH interventions for VYAs.

| Table 2 | Eligibility of the research question |
| --- | --- |
| **PCC element** | **Definition** |
| Population | The population for this study will be all adolescents from ages 10 to 19. |
| Concept | All sexual health interventions targeting adolescents published between 2000 and 2022. |
| Context | Research articles and publications are limited to sub-Saharan Africa. |

---

| Box 1   Sample data extraction table |
| --- |
| 1. Authors. |
| 2. Title. |
| 3. Year of publication. |
| 4. Country of origin. |
| 5. Aims of the study. |
| 6. Sample size. |
| 7. Methodology. |
| 8. Intervention. |
| 9. Outcomes measured. |
| 10. Duration of the study. |
| 11. Key findings that match the review questions. |

4. What are the gaps identified in these interventions?
5. What future programmatic work can be recommended on SH for VYAs in SSA?

### Stage 2: identifying relevant studies

To identify relevant studies for this review, a search for relevant articles and publications on adolescent sexual and reproductive health interventions in SSA will be conducted using several electronic databases including the Popline, EMBASE, PubMed, CINAHL, Dimensions, African Journals Online (AJOL) and summoncountry-specific search. Additional searches of key terms in Google Scholar and Directory of Open Access Journals (DOAJ) will be performed to find additional literature.

Several search techniques will be employed, including phrase searching, Boolean operators (AND, OR, NOT), Truncation, Wildcats, Field search, Proximity search, citation chaining and filters to limit searches to inclusion criteria presented in this protocol. The final search strings will be presented in the report (refer to online supplemental file 1). The search terms will include a combination of keywords and alternative terms such as: "Adolescent" OR "Very young Adolescents" OR "VYAs" OR "early adolescents" OR "Teenagers" OR "boys" OR "girls", AND "Sexual and Reproductive health" OR "Reproductive health" OR "sexual health" OR "reproductive health care" OR "contraception" OR teen pregnancy" OR "sex education", AND "Interventions" OR "Programming", AND "Sub-Saharan Africa" OR "Country name of a Sub-Saharan country". Using table 1, all researchers involved in the review will keep an up-to-date log of the number of articles found and the date found during each session of the literature search:

An academic librarian will be consulted to assist with identifying appropriate subject headings terms for the search and how to modify them for different databases that will be used. Relevant articles will be downloaded from the databases and uploaded into Mendeley (for importation and duplicates screening) first, then exported to Rayyan systematic review software for further screening of titles and abstracts.

### Stage 3: study selection

According to the Arksey and O'Malley framework, the third stage of the process of the scoping review is to select studies to be included in the review (see figure 1 in the additional files section after the references). The first step in the third stage of the process will involve the screening of the titles and abstracts of the peer-reviewed articles and publications that were downloaded in the second stage. The articles and publications will be screened according to the criteria that were used in the second stage, which will include only studies from SSA; only adolescent sexual and reproductive health interventions and all interventions that include adolescents. The second step in this process will be a review of full-text articles and publications whose titles and abstracts have been deemed relevant for this review and the screening will be done by two reviewers. Should there be a disagreement between the two reviewers, the reviewers will consult with a third reviewer who will decide on whether to include the article/publication or not.

### Inclusion criteria

The focus of the review will be peer-reviewed articles and publications from the SSA from 2003 to 2022. Other criteria for inclusion will include language (only English articles and publications), type of intervention (only SH and reproductive interventions) and study population (adolescents ages 10–19). These criteria were informed by the Population–Concept–Context framework (see table 2) recommended by the Joanna Briggs Institute for scoping reviews to determine the eligibility of research and is recommended as less restrictive alternative to the Population, Intervention, Comparator and Outcome mnemonic recommended for systematic reviews. Furthermore, we will include programmatic and intervention literature that has not been published in peer-reviewed articles.

### Exclusion criteria

The peer-reviewed articles and publications from outside the SSA, publications not in English, including publications before 2003 and after 2022 will not be included in this review.

### Stage 4: Charting the data

The fourth stage of the process of this scoping review will involve charting the data using Preferred Reporting Items for Systematic Review and Meta-Analysis Extension for Scoping review. This will involve the extraction of study characteristics and putting this information in a table or a chart. The study characteristics that will be extracted will include author, title, year of publication, study location, type of intervention, duration of intervention, study population and key findings (see box 1). The chart developed will be piloted by two reviewers separately to make sure that the chart is capturing information accurately. The whole process of extracting data from the studies will be done independently by two reviewers. To guarantee consistency between the reviewers, the charted data from

the two will then be compared, and any inconsistencies will be further explored to reach a consensus.

## Stage 5: collating, summarising and reporting the result

This stage will involve collating, summarising and reporting the results from the process in stage 4. This will provide summary information on what types of adolescent SRH interventions have been implemented for VYAs in SSA, the settings in which they were implemented, their effectiveness and implementation challenges and facilitators. In addition, this process will also show SH interventions that have been under-researched and may require further investigation.

## Stage 6: consultation exercise

At this stage, we will involve relevant stakeholders working in the field of Adolescent SH in Malawi. The stakeholders will be engaged to provide valuable insight beyond our scoping review report. Since these stakeholders will be experts working in the field of adolescent SH we envisage that they will add value to our work through providing additional references of SH interventions that the scoping review may have missed, suggesting existing SH interventions that have not been published and giving insights into our findings. We will organise an in-person workshop in Malawi where we will present our findings to stakeholders and allow them to contribute their insights into our work.

## ETHICS AND DISSEMINATION

Since the scoping review methodology involves putting together information from articles that are publicly available, this study does not require ethical approval. To disseminate our findings, we will submit a manuscript for publication in a scientific journal. Furthermore, the findings of this scoping review will be presented in various scientific fora and conferences. It is anticipated that the scoping review will provide a comprehensive overview of the evidence base of adolescent SH interventions for VYAs and will highlight areas where evidence is inconclusive or missing. The review will also provide key information to policymakers and health professionals interested in planning, funding and delivering evidence-based and effective SH for VYAs.

## Patient and public involvement

No patient will be involved.

## DISCUSSION

The proposed scoping review will generate a synthesis of SH programmes or interventions implemented that included VYAs in SSA. For a long time, SH programmes have focused on older adolescents aged 15–19 years overlooking VYAs.[24 27] As such, literature is replete with programmes or interventions for older adolescents.[8] As efforts are being put into positive youth development to promote adolescent SH,[27] attention is shifting to VYAs, resulting in an increase in both VYAs SH research and

programming over the past 5 years, globally.[2 20 23 36 41 42] However, there are very few SH interventions implemented among VYAs that have been documented in SSA.[2 27]

Findings from this scoping review will be relevant for both policymakers and SH intervention programmers for VYAs in SSA. For instance, the synthesis of SH interventions that will be documented will showcase what SH interventions have been implemented for VYAs so far in SSA, including the areas of focus, what works and what does not work in implementing these interventions. This knowledge is critical for developing evidence-based context-specific and culturally appropriate interventions for VYAs.

Furthermore, findings from this scoping review will be important for researchers working in the area of VYA SH as this scoping review may generate questions for further research around why some interventions work well and while others do not. In addition, the scoping review will contribute to knowledge by identifying gaps in SH interventions implemented for VYAs in SSA.

**Author affiliations**
[1]Public Health, Kamuzu University of Health Sciences, Blantyre, Malawi
[2]Public Health, University of the Witwatersrand Johannesburg, Johannesburg, South Africa
[3]Gender & Health Research Unit, South African Medical Research Council, Pretoria, South Africa
[4]School of Public Health, University of the Witwatersrand, Johannesburg, South Africa
[5]Obstetrics and Gynecology, University of Benin, Benin City, UK
[6]Applied Sciences, Malawi University of Business and Applied Sciences, Blantyre, Malawi

**Contributors** WCM conceived of the idea, developed the research questions and study methods and contributed meaningfully to the drafting and editing. BNK-K, AS and SJ helped in developing the research questions and study methods, contributed meaningfully to the drafting and editing. YS and MK supervised the preparation of the protocol and critically reviewed the manuscript for important intellectual content. All authors approved the final manuscript.

**Funding** This research (or '[initials of fellow]') will be supported by the Consortium for Advanced Research Training in Africa (CARTA). CARTA is jointly led by the African Population and Health Research Center and the University of the Witwatersrand and funded by the Carnegie Corporation of New York (Grant Number G-19-57145), Sida (Grant Number 54100113), Uppsala Monitoring Center, Norwegian Agency for Development Cooperation (Norad), and by the Wellcome Trust [reference number 107768/Z/15/Z] and the UK Foreign, Commonwealth & Development Office, with support from the Developing Excellence in Leadership, Training and Science in Africa (DELTAS Africa) programme. The statements made and views expressed are solely the responsibility of the Fellow. For the purpose of open access, the author has applied a CC BY public copyright licence to any Author Accepted Manuscript version arising from this submission.

**Competing interests** None declared.

**Patient and public involvement** Patients and/or the public were not involved in the design, or conduct, or reporting, or dissemination plans of this research.

**Patient consent for publication** Not applicable.

**Provenance and peer review** Not commissioned; externally peer reviewed.

**Data availability statement** All data will be published in a data repository once collected and can be accessed from the repository.

responsibility arising from any reliance placed on the content. Where the content includes any translated material, BMJ does not warrant the accuracy and reliability of the translations (including but not limited to local regulations, clinical guidelines, terminology, drug names and drug dosages), and is not responsible for any error and/or omissions arising from translation and adaptation or otherwise.

**ORCID iD**
Wanangwa Chimwaza Manda http://orcid.org/0000-0001-8061-8457

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
