## [Reviewer comments · BMJ Open]

ARTICLE DETAILS

TITLE (PROVISIONAL)	Adolescent Sexual health interventions that include very young adolescents in sub-Saharan Africa: A scoping review protocol
AUTHORS	Manda, Wanangwa; Sikweyiya, Yandisa; Kaunda-Khangamwa, Blessings; Selemani, Apatso; Jimu, Scholastica; Kamndaya, Mphatso

VERSION 1 – REVIEW

REVIEWER	Bhana, Deevia University of KwaZulu-Natal
REVIEW RETURNED	15-Jun-2022

GENERAL COMMENTS	For too long VYA have evaded the focus of attention across the globe but quite strikingly in SSA. Understanding how sexual health needs manifest for VYA is vital as is developing locally specific interventions that address these needs. This scoping review clearly spells out the gaps and its aim and intention. I support the publication with two minor considerations: 1. Please clarify: Most of these challenges are related to their sexual health (SH) most of which arise because of sexual risk behaviours in adolescence (11–17). Firstly this sentence requires some editing and clarity in relation to sexual health risks and gender norms. Gender is clearly missing in this framing as is a demonstration of structural and other social variables that impact on VYA experiences of sexuality and the implication for sexual health 2. What is the existing research on VYA and sexuality in SSA? I think it is important to frame this study against this available literature which highlights some of these issues In South Africa Bhana's work is necessary to cite as is a new piece of writing in Gender and Education 2022 (Catriona Macleod).
---

REVIEWER	Blum, Robert Johns Hopkins University
REVIEW RETURNED	05-Jul-2022

GENERAL COMMENTS	First, and most centrally, I do not understand why this is a manuscript that would be considered for publication in BMJ (or elsewhere). The authors simply explain what they will do using a fairly standard approach for scoping exercises. Having just finished one such review myself, I would have been hard pressed to think that journal readers would be interested in "what we will do". Rather, I think the interest would be in what was found.
--

	I have a few other concerns as well. The authors' central interest relates to programs and interventions with adolescents (including young adolescents). But they plan to limit their review to the peer reviewed literature. Why? Most of the programmatic and intervention literature is not found in peer reviewed articles. Let me try to answer some of the questions asked in the review sheet. I can answer some of the questions 1-15 in the affirmative but in truth, that says little about whether this is a publishable manuscript. 8. Are the references up-to-date and appropriate? some of the references are quite old. 12. Are the study limitations discussed adequately? No. As I note above, using only peer reviewed literature is a significant limitation. 14. To the best of your knowledge is the paper free from concerns over publication ethics (e.g. plagiarism, redundant publication, undeclared conflicts of interest)? I suspect that the answer will be yes but again I can't say given what I read.
--	---

VERSION 1 – AUTHOR RESPONSE

Comments from the First Reviewer		
Comment	Description	Response
1.	Please clarify: Most of these challenges are related to their sexual health (SH) most of which arise because of sexual risk behaviors in adolescence (11–17). Firstly this sentence requires some editing and clarity in relation to sexual health risks and gender norms. Gender is clearly missing in this framing as is a demonstration of structural and other social variables that impact on VYA experiences of sexuality and the implication for sexual health	Thank you for this important observation and suggestion. We have edited the sentence and provided the linkage between sexual health risks and gender norms, social economic, and structural factors. Please refer to the background section, third paragraph. We have highlighted the correction in green.
2.	What is the existing research on VYA and sexuality in SSA? I think it is important to frame this study against this available literature which highlights some of these issues. In South Africa Bhana's work is necessary to cite as is a new piece of writing in Gender and Education 2022 (Catriona Macleod).	Thank you for this important question. We have added some literature on VYA and sexuality in SSA. Please refer to the second paragraph in the background section. We have highlighted this section in light blue. We have also cited Bhana's and Macleod's work in the third paragraph sentence line number 4.
Comments from the Second Reviewer		

1.	First, and most centrally, I do not understand why this is a manuscript that would be considered for publication in BMJ (or elsewhere). The authors simply explain what they will do using a fairly standard approach for scoping exercises. Having just finished one such review myself, I would have been hard pressed to think that journal readers would be interested in "what we will do". Rather, I think the interest would be in what was found	We thank the reviewer for this comment. We wish to point out to the reviewer this is a scoping review protocol, not the actual scoping review. Scoping review protocols have been published in numerous journals including BMJ open. Scoping review protocols are important as they enable authors to have their scoping review aims, objectives, and methodology, including literature search criteria, peer-reviewed before they do the actual scoping review. Below we provide references to published Scoping Review protocols ^{1,2}. We plan to do the actual scoping review after the protocol has been peer-reviewed and accepted for publication. Reference 1. Sikweyiya Y, Nduna M, Khuzwayo N, Mthombeni A. Gender-based violence and absent fathers : a scoping review protocol. BMJ Open. Published online 2016:4-9. doi:10.1136/bmjopen-2015-010154 2. Bour C, Schmitz S, Ahne A, Perchoux C, Dessenne C, Fagherazzi G. Scoping review protocol on the use of social media for health research purposes. BMJ Open. Published online 2021:1-4. doi:10.1136/bmjopen-2020-040671
2.	I have a few other concerns as well. The authors' central interest relates to programs and interventions with adolescents (including young adolescents). But they plan to limit their review to peer-reviewed literature. Why? Most of the programmatic and intervention literature are not found in peer-reviewed articles.	Thank you for this important comment and observation. The authors acknowledged this as a limitation as well in the protocol. As per the reviewer's suggestion, we have now included 'programmatic and intervention literature that is not published in peer-reviewed articles. We have highlighted these changes in yellow (please refer to the abstract in methods and analysis section; and in the study selection section in the inclusion criteria on page 9). In addition, the authors also included an additionastep in the protocol that states that stakeholders will be consulted and engaged to provide insights on interventions that may not have been published or missed by the scoping review.

3.	8. Are the references up-to-date and appropriate? some of the references are quite old.	Thank you for this important observation. We have removed old references, and now only included those between 2003 to 2022.
4.	12. Are the study limitations discussed adequately? No. As I noted above, using only peer-review literature is a significant limitation.	Thank you for this important comment and observation. The authors acknowledged this as a limitation as well in the protocol. As per the reviewer's suggestion, we will now include 'programmatic and intervention literature that is not published in peer-reviewed articles. Furthermore, we will also consult relevant stakeholders (e.g. programme implementers who work with adolescents on SH) provide insights on programmes or interventions which are not published or may have been missed by the scoping review.
5.	14. To the best of your knowledge is the paper free from concerns over publication ethics (e.g. plagiarism, redundant publication, undeclared conflicts of interest)? I suspect that the answer will be yes but again I can't say given what I read.	Thank you. We have submitted to Turnitin the protocol to check for plagiarism and can provide a report if required.

VERSION 2 – REVIEW

REVIEWER	Bhana, Deevia University of KwaZulu-Natal
REVIEW RETURNED	06-Sep-2022
GENERAL COMMENTS	Changes made to my satisfaction